# Gulf War Illness Induced Sex-Specific Transcriptional Differences Under Stressful Conditions

**DOI:** 10.3390/ijms26083610

**Published:** 2025-04-11

**Authors:** Joshua Frank, Lily Tehrani, Jackson Gamer, Derek J. Van Booven, Sarah Ballarin, Raquel Rossman, Abraham Edelstein, Sadhika Uppalati, Ana Reuthebuck, Fanny Collado, Nancy G. Klimas, Lubov Nathanson

**Affiliations:** 1Institute for Neuro-Immune Medicine, Dr. Kiran C. Patel College of Osteopathic Medicine, Nova Southeastern University, Fort Lauderdale, FL 33328, USA; jf2048@mynsu.nova.edu (J.F.); fcollado1@nova.edu (F.C.); nklimas@nova.edu (N.G.K.); 2Dr. Kiran C. Patel College of Osteopathic Medicine, Nova Southeastern University, Fort Lauderdale, FL 33328, USA; lt1125@mynsu.nova.edu (L.T.); jg3380@mynsu.nova.edu (J.G.); sb3290@mynsu.nova.edu (S.B.); rr1800@mynsu.nova.edu (R.R.); ae1079@mynsu.nova.edu (A.E.); 3John P. Hussman Institute for Human Genomics, Miller School of Medicine, University of Miami, Miami, FL 33146, USA; dvanbooven@med.miami.edu; 4Halmos College of Arts and Sciences, Nova Southeastern University, Fort Lauderdale, FL 33328, USA; su106@mynsu.nova.edu (S.U.); ar3019@mynsu.nova.edu (A.R.); 5Department of Veterans Affairs, Miami VA Healthcare System, Geriatric Research Education and Clinical Center (GRECC), Miami, FL 33125, USA

**Keywords:** Gulf War Illness (GWI), transcriptomics, sex differences, peripheral blood mononuclear cells, PBMCs, RNA sequencing, RNA-seq, immune dysregulation, oxidative stress, inflammation, cytokine signaling

## Abstract

Gulf War Illness (GWI) is a multi-symptom disorder affecting 1990–1991 Persian Gulf War veterans and is characterized by post-exertional malaise, neurological symptoms, immune deregulation, and exhaustion. Causation is not understood, and effective diagnostics and therapies have not yet been developed. In this work, we analyzed stress-related, sex-specific transcriptomic shifts in GWI subjects and healthy controls through RNA sequencing of peripheral blood mononuclear cells (PBMCs). Blood samples at baseline (T0), at maximal exertion (T1), and four hours post-exertion (T2) were analyzed. In female subjects with GWI, pathways associated with pro-inflammatory processes were found to be deregulated, and in male GWI subjects, pathways related to IL-12 signaling and lymphocytic activation were deregulated at T1 compared to T0. During recovery from stress, pathways corresponding to immune responses and microglial cell activation were altered in female GWI subjects, and apoptotic signaling changed in males with GWI. Documented sex-specific immune deregulation leads to finding better biomarkers. Targeting sex-specific transcriptomic markers of the disease could lead to new therapies for GWI.

## 1. Introduction

Gulf War Illness (GWI) is a multi-symptom, complex, and debilitating disease, which has been reported by the veterans who were deployed in the 1990–1991 Persian Gulf War [1,2]. Veterans with GWI commonly present with fatigue, neurological dysfunction, neuromuscular and joint pain, symptoms related to the gastrointestinal tract, impaired immune functions, sensitivity to odors, chemicals, and light, post-exertional malaise, and other related symptoms that were reported during or immediately after the Gulf War [3,4,5,6]. It is suggested that GWI is caused by a combination of factors such as toxic exposure and modifications of epigenetic profiles [7,8,9].

Some advances have been made to determine the underlying mechanisms of GWI. Haley et al. found that exposure to sarin nerve gas, combined with the specific genetic mutation, is likely responsible for GWI [10]. Golomb et al. highlighted two leading hypotheses for major mechanisms underlying GWI: inflammation and mitochondrial impairment [11]. Steele et al. concluded that a limited number of wartime exposures played a role in the etiology of GWI but these factors differed in importance depending on where veterans served [12]. However, the exact causes of GWI remain unclear [10] and no effective diagnostic approaches and/or treatment modalities have been made available to date [13,14,15].

While the underlying mechanisms of GWI are not fully elucidated, research efforts have highlighted some novel approaches to better understand the pathophysiology. As part of our ongoing research efforts, we carried out transcriptomic profiling of circulating immune cells from male GWI patients and healthy controls (HCs) to gain insights into the gene expression dysregulation tied to GWI and stress [16]. Using an exercise challenge protocol, we collected blood samples from male GWI and HC subjects at baseline (T0) before the stress event (exercise), at the point of maximal exertion (T1), and four hours after recovery from maximal exertion (T2). Peripheral blood mononuclear cells (PBMCs) were used to extract RNA and perform transcriptomics profiling using RNA sequencing (RNA-seq) at each time point. Our data identified dysregulation in immune and inflammatory pathways and novel immune and inflammatory markers in GWI male patients compared to HCs. Particularly, differential expression of the genes involved in immune system processes, the inflammatory response, cytokine signaling pathways, T-cell activation and differentiation, and NF-κB signaling was found to be dysregulated in GWI patients compared to healthy controls, both at baseline and in response to the exercise challenge [16].

In the current study, we extended our exercise challenge strategy to both male and female GWI patients and HCs to identify sex-dependent transcriptional changes in response to stress in GWI. We performed RNA-seq on PBMCs from female and male GWI patients and HCs under the same modeled stress conditions to evaluate transcriptomic alterations and affected molecular pathways. We identified sex-dependent pathways at maximal exertion (such as regulation of leukocyte chemotaxis, oxidative stress pathways, leukocyte activation response to a toxic substance, IL-12, and pro-inflammatory and profibrotic mediators) and at recovery from maximal exertion (such as human cytomegalovirus (HCMV) or human herpes virus 5 pathways (HHV-5), oxidative stress-induced senescence, the regulation of microglial cell activation, negative regulation of the extrinsic apoptotic signaling pathway, and negative regulation of type II interferon production).

Our findings highlight the sex-dependent nature of GWI pathogenesis, which could be used to more effectively diagnose and treat GWI patients. It is our overarching goal to target treatment more effectively, outside symptom amelioration, and identify therapeutic targets for GWI.

## 2. Results

### 2.1. Participant Characteristics

In total, 25 female and 19 male GWI subjects and 18 female and 20 male age- and body mass index (BMI)-matched HCs participated in this study. We found no significant differences based on age and BMI between male and female GWI subjects or male and female HCs (Table 1). Levels of mental and physical disability were evaluated via the Short Form 36-Item Survey (SF-36) questionnaire [17]. Scores were evaluated on a 100-point scale with lower scores indicating greater levels of disability. Female GWI subjects showed significant differences (*p* < 0.05) in physical and mental health when compared to HCs. We have previously published on the effects of GWI on the levels of mental and physical disability of male patients [16].

### 2.2. Transcriptomic Changes Between Maximum Exertion (T1) and Baseline Before Exercise Challenge (T0) Stratified by Sex

Gene expression in the PBMCs of both GWI subjects and HCs was compared at T1 (maximal exertion) versus T0 (baseline). These results were stratified by sex and differentially expressed genes (DEG) were analyzed using Metascape [18] for the functional comparative analysis.

Female GWI subjects had twenty-seven significant DEGs at T1 vs. T0 (False Discovery Rate (FDR) < 0.1), six of which were underexpressed and twenty-one of which were overexpressed (Appendix A). When comparing female HCs at T1 vs. T0, significant (FDR < 0.1) differential expression in 60 genes was found. Of these sixty genes, fifty-nine were overexpressed and one was underexpressed (Appendix A). DEGs in male HCs and GWI subjects were compared in our previous publication [16].

To identify sex differences in affected pathways and ontologies in both GWI and HC cohorts in response to the maximal exertion, DEGs were analyzed through comparative functional analysis of gene networks using Metascape [18]. Specifically, we focused on the functional groups related to immune responses, detoxification, and oxidative stress because these processes have been shown to be disrupted in GWI models [19]. We compared the most affected pathways and ontologies between male and female cohorts (Figure 1A–D and Figure 2A).

In female GWI patients compared to males, pathways of high significance included the pro-inflammatory and profibrotic mediators, regulation of leukocyte chemotaxis, response to toxic substances, and secretion by cells (Figure 2A). In male GWI patients, unlike females, pathways that were observed to have high significance included the interleukin IL-12 pathway, natural killer (NK)-mediated cytotoxicity, immunoregulatory interactions between a lymphoid and non-lymphoid cell, regulation of lymphocyte-mediated immunity, regulation of leukocyte-mediated immunity, lymphocyte activation, regulation of NK cell-mediated immunity, and positive regulation of NK cell-mediated cytotoxicity (Figure 2A).

Of note, both female and male GWI patients demonstrated significant changes to various degrees in cell activation, leukocyte activation, positive regulation of immune effector process, positive regulation of the defense response, innate immune response, IL-10 signaling, cell killing, and regulation of leukocyte chemotaxis (Figure 2A).

Female HCs, unlike their healthy male counterparts, demonstrated highly significant (FDR < 0.1) changes in regulated exocytosis, cellular response to oxidative stress, and secretion by cells (Figure 2A). At the same time, in male HCs, we found highly significant positive regulation of NK cell-mediated cytotoxicity, development and heterogeneity of the innate lymphoid cell (ILC) family, immune response-activating signaling pathway, immune response-activating cell surface receptor signaling pathway, regulation of chemotaxis, and viral protein interaction with cytokine and cytokine receptors (Figure 2A).

Both female and male HCs were found to have significant changes to various degrees in pathways of negative regulation of leukocyte activation, immunoregulatory interactions between a lymphoid and a non-lymphoid cell, and the primary interaction database (PID) IL-12 pathway (Figure 1C,D and Figure 2A).

### 2.3. Transcriptomic Changes Between T2 (4 h After Maximal Exertion) and T1 (Maximal Exertion)

Transcriptomic changes in PBMCs were compared in GWI patients and HCs between recovery from the exercise challenge (T2) and maximal exertion (T1) time points. The results were stratified by sex and functional comparative analysis, which was performed using Metascape [18].

Female GWI subjects had 675 DEGs between T2 and T1, 442 of which were underexpressed and 232 were overexpressed (Appendix A). However, when comparing gene expression in female HCs at T2 vs. T1, 585 DEGs were found. Of these 584 DEGs, 232 were overexpressed and 352 were underexpressed (Appendix A). Results of differential gene expression in male GWI patients and HCs have been published previously [16].

Functional pathway analysis conducted on DEGs between male and female GWI cohorts at T2 and T1 demonstrated significant changes between the sexes (Figure 3A,B and Figure 4A). In the male GWI cohort, unlike females, pathways of high significance included those associated with the mucosal immune response, tissue-specific immune response, antibacterial humoral response, immunoregulatory interactions between a lymphoid and a non-lymphoid cell, regulation of NK cell-mediated immunity and cytotoxicity, positive regulation of IL-12 production, and positive regulation of the intrinsic apoptotic signaling pathway (Figure 4A).

In the female GWI cohort compared to males, pathways of high significance included those associated with negative regulation of type II interferon production, leukocyte differentiation, regulation of neuroinflammatory response, regulation of microglial cell activation, regulation of inflammatory response, negative regulation of the extrinsic apoptotic signaling pathway, and negative regulation of the oxidative stress-induced senescence pathway (Figure 4A).

In addition, pathways of high significance to various degrees in both male and female GWI patients included those tied to systemic lupus erythematosus, activated *PKN1*, HCMV events, activation of HOX genes during differentiation, oxidative stress-induced senescence, lymphocyte activation, and immune response-activating signaling pathways (Figure 4A).

Regulation of the neuroinflammatory response was highly significant in female HCs, unlike their male HC counterparts (Figure 4A). We also observed that in both male and female HCs, pathways of high significance included those tied to immunoregulatory interactions between a lymphoid and a non-lymphoid cell, regulation of NK cell-mediated immunity and cytotoxicity, lymphocyte activation and differentiation, immune response-activating signaling pathway, regulation of microglial cell activation, regulation of extrinsic apoptotic signaling pathway via death domain receptors, and regulation of intrinsic apoptotic signaling pathways (Figure 3C,D and Figure 4A).

### 2.4. Nanostring Validation

We performed validation of RNA-seq results using Nanostring technology. This technology involves neither amplification bias nor conversion of RNA into cDNA. Nanostring utilizes probes that are complementary to the target transcripts identified in RNA-seq and have an attached specific combination of six fluorophores. Nanostring counts these distinct fluorophores’ combinations.

We considered the differential expression of the gene to be validated only if both RNA-seq and Nanostring demonstrated the same expression pattern, either overexpression or underexpression. A total of 95 percent of the DEGs identified by RNA-Seq were validated by Nanostring technology (Appendix A). Genes with discrepancies between the Nanostring and RNA-seq were not used in the functional analysis.

## 3. Discussion

This project was the first effort to analyze sex differences in the transcriptomes of circulating immune cells among GWI patients under stress. Specifically, we evaluated transcriptomic changes provoked by exercise challenge: first, between the point of maximal exertion (T1) and baseline (T0); and second, between recovery (T2) and the point of maximal exertion (T1). Our results show that biological sex is likely to influence gene expression of PBMCs in GWI.

### 3.1. Transcriptomic Changes Between Maximal Exertion (T1) and Baseline Before Exercise (T0) in GWI Patients and HCs

We found significant changes in response to the exercise challenge in the PBMCs of all participants. Functional analysis of the DEGs revealed a broad number of affected pathways.

Functional analysis of the DEGs in male GWI patients, compared to their female GWI counterparts, in response to maximal exertion (T1 vs. T0) showed that positive regulation of NK cell-mediated cytotoxicity and regulation of leukocyte chemotaxis were highly affected. These pathways were also significantly affected in the HCs.

#### 3.1.1. Regulation of Leukocyte Chemotaxis

We found that gene networks tied to the regulation of leukocyte chemotaxis were of significance among female GWI (Figure 2A,B) and male HC subjects (Figure 1C,D and Figure 2A), unlike their male GWI and female HC counterparts. This could allude to potential underlying sex-dependent regulatory mechanisms in GWI. Pro-inflammatory cytokines were found to be increased in GWI veterans, contributing to the dysregulation of leukocyte chemotaxis from a cytokine imbalance [20]. In addition, in patients with Myalgic Encephalomyelitis/Chronic Fatigue Syndrome (ME/CFS)—a disease that shares many symptoms with GWI—pro-chemotactic factors are significantly deregulated [21,22,23]. Given what has been published, this could indicate the potential for a biomarker of GWI in female patients. The absence of an effect on this pathway in male GWI patients may require more research.

#### 3.1.2. Oxidative Stress

We observed that pathways related to the cellular response to oxidative stress are uniquely activated in female HCs compared to males; however, changes in these pathways are not significant enough in GWI subjects (Figure 2A). The absence of changes in oxidative stress upon exertion in GWI cohorts could indicate that at baseline, GWI patients are already experiencing a level of exertion similar to the T1 time point of HCs.

Oxidative stress pathways were found to be significantly deregulated in the rat model of GWI. In addition, malondialdehyde (MDA), a by-product of peroxidation, is an oxidative stress marker that was also found to be increased in rat models of GWI [19]. Furthermore, stress studies in ME/CFS patients also found increased markers of oxidative stress at rest that further rose after exercise [24].

#### 3.1.3. Response to a Toxic Substance

Response to a toxic substance was shown to be uniquely deregulated in male GWI subjects compared to females (Figure 2A). It has been well-established that Gulf War veterans were exposed to various environmental toxins [25]. To reduce cellular damage from chemical exposures, redox balance pathways are utilized [26]. Considering that female GWI subjects have deregulated changes in oxidative stress pathways (see above), this could hint at a sex-mediated mechanism in which GWI females are unable to maintain redox balance and detoxification processes.

#### 3.1.4. Immune Dysfunction

Functional analysis of the DEGs showed that the negative regulation of leukocyte activation, immunoregulatory interactions between a lymphoid and a non-lymphoid cell, regulation of NK cell-mediated cytotoxicity, and the PID IL-12 pathway were significantly enriched in both HC male and female subjects and in male GWI subjects but not among female GWI subjects (Figure 2A,B). This could suggest that there is a dysregulation of these functions in female GWI subjects.

#### 3.1.5. Leukocyte Activation and Lymphoid/Non-Lymphoid

Leukocyte activation and lymphoid/non-lymphoid cell pathways are significantly changed between T1 and T0 in all cohorts, except for in GWI females (Figure 2A,B). Although regulation of the leukocyte activation pathway is not fully understood in GWI patients, research suggests that factors such as immune system dysregulation can contribute to negative leukocyte regulation. Specifically, it was found that the immune dysfunction in GWI subjects results from decreased NK cytotoxicity, immune cell impairment, changes in the levels of pro-inflammatory cytokines, and cytokine signaling [27]. Together, these immune changes can cause defects in the immune cell signaling of regulatory pathways that serve to dampen the immune response, contributing to higher levels of inflammation [27]. Dysregulated communication between lymphoid and non-lymphoid cells was observed when analyzing the innate immune system among GWI subjects [28]. The pathophysiology of ME/CFS is also associated with dysregulation of the innate immune system [28]. Despite the overall increased levels of inflammation seen in ME/CFS, aspects of negative regulation of leukocyte activation are also present [21,29].

#### 3.1.6. IL-12

IL-12 pathways are changed between T1 and T0 in all cohorts, except for in GWI females (Figure 1A–D and Figure 2A,B). IL-12 is a pro-inflammatory cytokine that participates in various immune functions and responses, particularly in the antigen presentation and differentiation of naive T cells. These cytokines also regulate the functions of various effector cells and are often therapeutic targets in inflammatory and autoimmune diseases [30]. Prior studies have shown that the IL-12 response demonstrates significant sex-specific differences. Male GWI subjects showed significant alterations to the IL-12 signaling pathway. IL-12 stimulates the proliferation and activation of T cells and NK cells [31,32] and the release of IFN-y [32]. IFN-y has also been characterized as a pro-apoptotic factor [33], which may be related to the dysregulated intrinsic apoptotic signaling pathway in male GWI patients (cell killing in Figure 2A). Patients with ME/CFS were also found to have dysregulation in the IL-12 pathway, most prominently in males [34]. Though further study is warranted, it has been found that in ME/CFS patients who were exposed to toxic substances, greater levels of immune dysregulation have been observed [35].

#### 3.1.7. Pro-Inflammatory and Profibrotic Mediators

Only in the female GWI cohort were pathways related to pro-inflammatory and profibrotic mediators enriched (Figure 2A). This enrichment could be associated with immune responses among female GWI patients. Previous studies have found that estrogen may inhibit cytokine storm [36]. This hints at a potential sex hormone-mediated mechanism that results in a deregulated response to pro-inflammatory and profibrotic mediators in female GWI subjects (Figure 2A). Furthermore, the recent literature on GWI has shown that patients with GWI were found to have increased levels of pro-inflammatory cytokines [32,37]. It was also published that the pathophysiology of ME/CFS is related to the activation of multiple immune–inflammatory pathways [38].

#### 3.1.8. Overall Transcriptomic Changes Between Maximal Exertion (T1) and Baseline (T0)

Based on these findings, GWI patients appear to exhibit multiple pathways contributing to their pathological response to stress. Notably, genes related to toxic exposure and oxidative stress—key mechanisms for combating environmental toxins—are significantly (*p*-value < 0.01) altered in GWI patients during their initial stress response. These findings align with GWI patients’ documented exposure to hazardous substances such as organophosphates, pyridostigmine bromide, and combustion products and suggest that an inability of their body to adapt and or break down these hazards may be a key driver of the disease. Additionally, chronic immune system dysregulation likely plays a central role in their impaired stress response as genes associated with immune cell function, immune mediators, and signaling molecules show significant (*p*-value < 0.01) alterations.

### 3.2. Transcriptomic Changes Between Four Hours After Maximal Exertion (T2) and Maximal Exertion (T1) in GWI Patients and HCs

#### 3.2.1. HCMV Events

Functional analysis of DEGs in GWI subjects during recovery (T2 vs. T1) showed that HCMV events were found to be significantly enriched among female GWI subjects compared to female HCs (Figure 4B). Metascape showed that this pathway was associated with human histone proteins (Appendix A), indicating that enrichment of this pathway is not related to the expression of viral genes. Rather, it is part of the innate immune response as histones can act as Damage-Associated Molecular Patterns (DAMPs), promoting immune cell activation and pro-inflammatory cytokine release [39]. This confirms that there is dysregulation of the immune response in female GWI subjects. Additionally, it is worth noting that it was not observed in male GWI subjects. This could indicate a sex-specific response.

#### 3.2.2. Oxidative Stress-Induced Senescence

Oxidative stress-induced senescence occurs at high concentrations of reactive oxygen species (ROS) in a cell and contributes to cellular senescence through various downstream protein interactions [40,41]. The changes in this pathway during recovery among healthy females but not females with GWI (Figure 4B) suggest that it could be involved in sex-specific GWI onset and/or progression. The underlying mechanisms tied to the oxidative stress-induced senescence pathway associated with GWI have yet to be fully elucidated. Higher levels of oxidative stress can decrease antioxidant levels, promoting cellular damage and senescence [41,42].

#### 3.2.3. Leukocyte Differentiation

In addition, we found that leukocyte differentiation was significantly (*p*-value < 0.01) enriched in female GWI patients compared to male and female HCs in response to recovery (Figure 4A). As such, this pathway could provide a method for differentiating between GWI patients and HCs. Previous publications have observed that the differentiation of various immune cells is broadly inhibited by anti-estrogens, which could indicate that leukocyte differentiation is upregulated by high estrogen levels [43]. This could provide a reasonable explanation as to why leukocyte differentiation was enriched in female GWI subjects but not in their male counterparts. While leukocyte differentiation has not been explicitly studied in GWI, it may be assumed that pro-inflammatory cytokines excessively promote the differentiation of leukocytes [6].

#### 3.2.4. Regulation of Extrinsic Apoptotic Signaling

The regulation of the extrinsic apoptotic signaling pathway is significantly (*p*-value < 0.01) deregulated in GWI females (Figure 4A). It is unclear how this correlates to the progression of GWI; however, previous studies among ME/CFS patients have shown that reduced levels of adenosine triphosphate (ATP) likely contribute to the impairment of the extrinsic apoptotic pathway [44]. It is known that testosterone can increase levels of ATP [45], and male GWI subjects have been predicted to exhibit lower levels of testosterone [46]. This could provide a reasonable explanation for why male GWI subjects did not exhibit enrichment of this pathway.

#### 3.2.5. Negative Regulation of Type II Interferon

Negative regulation of type II interferon production was significantly (*p*-value < 0.01) changed in female GWI subjects compared to males with GWI in response to recovery but did not change in either male or female HC subjects (Figure 4A). Genes associated with this pathway were found to be significantly (FDR < 0.1) underexpressed (Appendix A, Figure 4A), which could indicate that the production of type II interferons is increased. This discovery could be of interest in further research into the biomarkers of GWI in female patients. Previous publications have shown that type II interferon levels were significantly increased (*p*-value < 0.01) in GWI patients [47]. Meanwhile, it remains unclear why females with GWI, but not males, experience increased production of type II interferons, highlighting the need for further research. Immunoregulatory interactions between a lymphoid and non-lymphoid cell, positive regulation of NK cell-mediated cytotoxicity, lymphocyte activation, and the immune response-activating signaling pathway were found to be similarly enriched at T2 vs. T1 (Figure 4A) as discussed in T1 vs. T0 (see Section 3.1).

#### 3.2.6. Overall Transcriptomic Changes Between Four Hours After Maximal Exertion (T2) and Maximal Exertion (T1)

These findings emphasize the significant sex-dependent changes in a variety of biological pathways that could underlie a failure of GWI patients’ ability to recover following various stressors. Sex-specific inflammation, oxidative stress, and deregulation of the immune system were detected. In both males and females, compromised cellular repair, differentiation, and immune processes could underlie the chronic nature of GWI. These findings point to the immediate need for personalized diagnostic markers and treatment strategies, accounting for sex-based immune and metabolic differences, to improve targeted interventions for GWI patients.

## 4. Materials and Methods

### 4.1. Cohort

This project was conducted in the Miami/Fort Lauderdale area and included 25 female subjects clinically diagnosed with GWI and 18 sedentary female HCs. It also included 19 male GWI subjects and 20 sedentary male HCs. All individuals with GWI and the HCs were recruited as part of a larger study. This study was conducted under the Declaration of Helsinki. Protocols were approved by the Nova Southeastern University (NSU)’s Institutional Review Board (2016-2-NSU). All subjects provided informed consent. GWI subjects that met the criteria for diagnosis (including the Centers for Disease Control and Prevention (CDC) [48] and Kansas criteria for GWI [1]) were included in this study. Subjects were excluded if they presented with a history of active smoking or alcohol misuse, diabetes, immunodeficiency, cardiovascular disease, stroke, autoimmunity, malignancy, or systemic infection present within two weeks of blood collection. Both cohorts were matched for age and BMI. Female subjects were also required to complete a gynecologic questionnaire to ensure that blood collection occurred during the first two weeks of their menstrual cycle.

SF-36 questionnaires [17] were used to evaluate the physical and mental health of GWI and HC subjects. SF-36 scores ranging from zero to one hundred were evaluated, where higher scores indicated lower levels of disability. SF-36 questionnaires were used to compare individuals with GWI and HCs including “Physical Health”, (which comprises eight domains of well-being including physical functioning, physical role functioning, bodily pain, and general health perception) as well as “Mental Health,” (which includes vitality, social function, emotional role, and mental health). Each of the eight listed domains was transformed into a 0-to-100-point scale, with each question being weighted equally. Higher scores on this scale indicated lower levels of disability. The SF-36 questionnaire includes eight scaled scores, which are weighted sums of the questions in their respective section. All scales were directly transformed into a 0-to-100 scale on the assumption that each question holds equal weight.

Participants were fed a standardized breakfast (yogurt and banana) and rested for 30 min in reclining chairs before blood draws (T0). Following this, a standard maximal graded exercise test (GXT) was conducted according to McArdle’s protocol [49]. This protocol was used as a part of a larger study to determine the biological mechanisms underlying neuroimmune diseases. GXTs involved participants pedaling at 60 W for two minutes, with an increase of 30 W every two minutes until maximal exertion was reached. A second blood draw was taken at the point of maximal exertion (T1), and a third blood draw was performed four hours after maximal exertion (T2).

### 4.2. PBMC Isolation and RNA Extraction

All Participants donated up to 8 mL of whole blood, which was then collected in K2EDTA tubes and diluted in a 1:1 ratio (*v*/*v*) in phosphate-buffered saline (PBS) within two hours. The resulting solution was layered with Ficoll-Paque Premium (GE Healthcare, Chicago, IL, USA). From there, samples were subjected to centrifugation at 500× *g* for 30 min at 20 °C without brakes. The PBMC layer was then washed with PBS and resuspended in a freezing medium. Aliquots of 10^7^ cells per mL were frozen in liquid nitrogen until preparations for analysis were completed. Total RNA was extracted using RNAzol (Molecular Research Center, Cincinnati, OH, USA), and quality was assessed using Agilent TapeStation 4200 (Agilent, Santa Clara, CA, USA), with all samples returning an RNA integrity number > 7.

### 4.3. RNA-Seq

PBMC samples of male GWI subjects and healthy controls were subsetted from a larger ongoing study. These data were described in our previously published manuscript [16]. To investigate sex differences in GWI pathology, we sequenced RNA from the PBMCs of female GWI subjects and HCs that underwent the same exercise challenge.

A total of 300 ng of RNA was submitted to the Genomic Core Facility of NSU and to the Center of Genome Technology (CGT) at the University of Miami for RNA-Seq. TruSeq Stranded Total RNA Library Prep Kit (Illumina Inc., San Diego, CA, USA) was used to generate libraries with a paired-end sequencing reading length of 150 base pairs. Illumina RNA-Seq pipeline was employed to determine genomic coverage, alignment percentage, and nucleotide quality for quality control testing. After quality control, raw sequencing data were converted to a fastq format.

### 4.4. RNA-Seq Analysis

GSNAP (version 2021-02-22) [50], HISAT2 (version 2.1.0) [51], and STAR (version 2.7.8a) [52] software were used to map raw reads to the GRCh38/release 103 reference genome. After alignment, analysis of the data produced by each aligner was conducted separately through batch effect mitigation and differential expression. Read counts aligned by GSNAP and HISAT2 were evaluated using HTSeq (version 0.13.5) [53], while STAR read counts were determined using the “--quantMode Transcriptome SAM” option. We combined gene counts from male GWI subjects and HCs from the previously published study [16] with gene counts from female GWI subjects and HCs. Genes from sex chromosomes were removed, and ComBat-seq [54] was utilized to adjust false positives generated from batch effects. Gene counts produced by each aligner were imported into Bioconductor/R (version 4.2.1) package DESeq2 (version 1.36.0) [55] for differential gene expression analysis. Genes with counts of 50 or higher in all samples in one or more groups (GWI patients at T0, GWI patients at T1, GWI patients at T2, HCs at T0, HCs at T1, HCs at T2) were used for differential analysis by DESeq2, resulting in a total of 14,986 genes for GSNAP, 11,794 genes for HISAT2, and 11,768 genes for STAR. DEGs were selected based on two criteria: (1) fold change (FC) > 1.5 and FDR < 0.1 in one out of three aligners, and FC > 1.4 and FDR < 0.15 in the remaining two aligners; or (2) FC > 1.5 and FDR < 0.10 in at least two out of three aligners.

Raw data were submitted to GEO with the accession number GSE286345.

After selection, DEGs were uploaded to Metascape [18]. Metascape identifies ontology terms that contain a statistically greater number of genes from the input list than expected by chance by employing the hypergeometric test for functional enrichment analysis. It calculates *p*-values using the hypergeometric distribution and controls FDR using the Benjamini–Hochberg algorithm for *p*-value correction. For the analysis, we applied the default Metascape parameters, which are a *p*-value threshold of <0.01, a minimum count of three genes from the input gene list in a pathway or ontology, and an enrichment factor of >1.5. Cytoscape (version 3.8.2) [56] was then employed to perform visualization of the GO networks.

### 4.5. Validation of RNA-Seq Results

A custom panel was used on the Nanostring *nCounter* platform to validate RNA-Seq findings. Hybridization and processing of 100 ng of total RNA in conjunction with counting was carried out according to manufacturer guidelines. Raw counts were analyzed using NanoString nSolver v.4 software. Geometric means of the negative controls plus two standard deviations were computed for all samples. Counts below the threshold value were removed from normalization and analysis. All procedures were executed following the manufacturer’s instructions.

## 5. Conclusions

In conclusion, our results illustrate sex-dependent transcriptomic differences in the effects of stressors on male and female GWI patients, which warrants further investigation. This was found to hold in both response to maximal exertion and recovery where the symptoms of GWI are known to be exacerbated [57]. It is important to note that our study findings have limitations. Specifically, larger-scale cohorts and reproduced data would allow for a comprehensive examination of transcriptomic profiles and confirm potential biomarkers that could be associated with disease progression under stress. Greater lengths of 48 h of exposure to exercise challenges could also yield valuable results. Furthermore, the inclusion of further multi-omics data in the same participants could provide a more encompassing understanding of the molecular mechanisms and pathways involved. As far as we know, this is the first and only study on sex differences at a transcriptomic level in GWI veterans. Thus, more studies directly on GWI veterans are needed to provide a valid and comprehensive understanding of GWI pathophysiology.

## Figures and Tables

**Figure 1 ijms-26-03610-f001:**
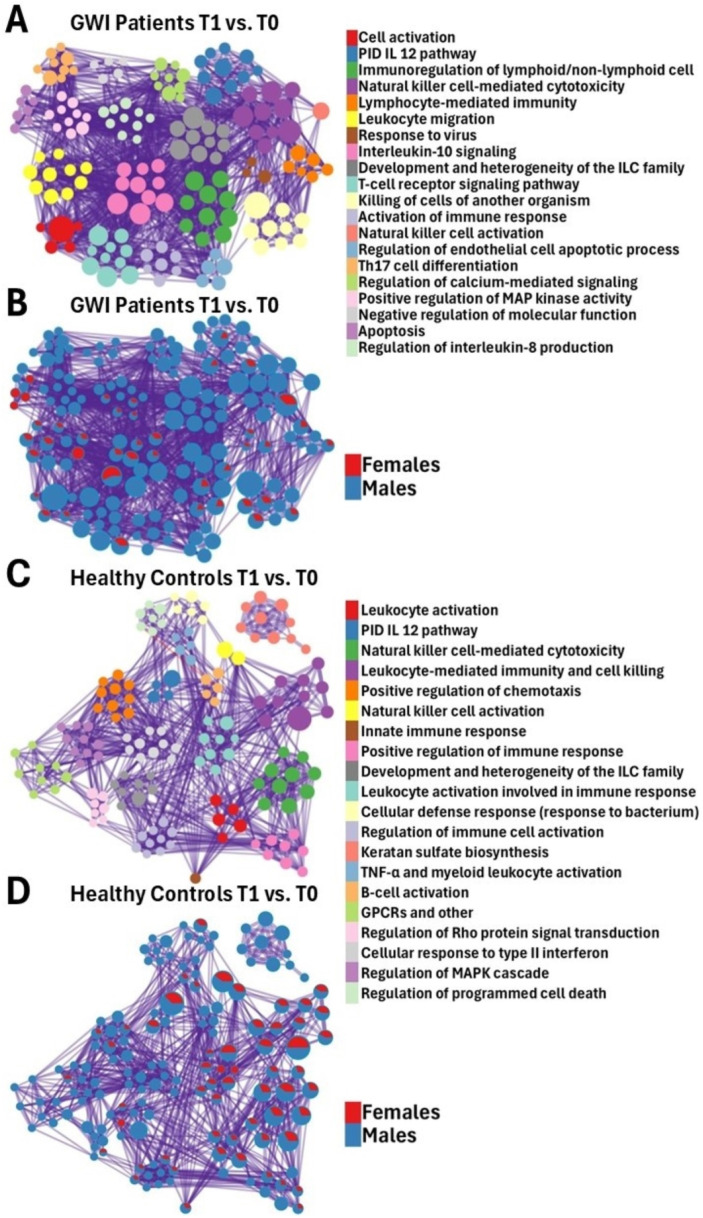
Metascape [18] expression analysis of DEGs between male and female GWI patients (**A**,**B**) and HCs (**C**,**D**) between T1 and T0. Each node represents an enriched term (**A**,**C**). The larger the representation of color, the more enriched it is in each cohort. Cutoff values included a *p*-value < 0.01, a minimum count of 3 in the list of differentially expressed genes, and an enrichment factor of >1.5. GO biological processes, KEGG pathways, Reactome gene sets, CORUM complexes, and canonical pathways from MSigDB were included in the search; (**B**,**D**) nodes are colored by ratios between male (blue) and female (red) participants. The size of the nodes is proportional to the z-score calculated by Metascape [18].

**Figure 2 ijms-26-03610-f002:**
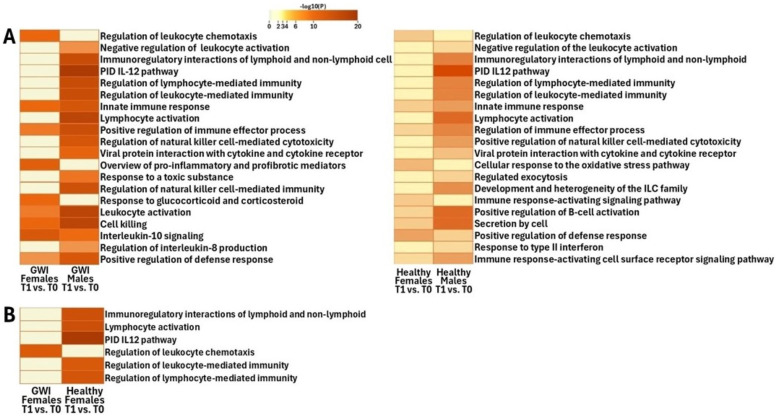
(**A**) Heatmap of the individual pathways that showed the largest difference between T1 and T0 in PBMCs of male and female participants that were found to match those pathways discovered in HCs or exhibited a change in which the cohort was affected compared to HCs. (**B**) A comparison of enriched pathways between female GWI veterans and healthy controls.

**Figure 3 ijms-26-03610-f003:**
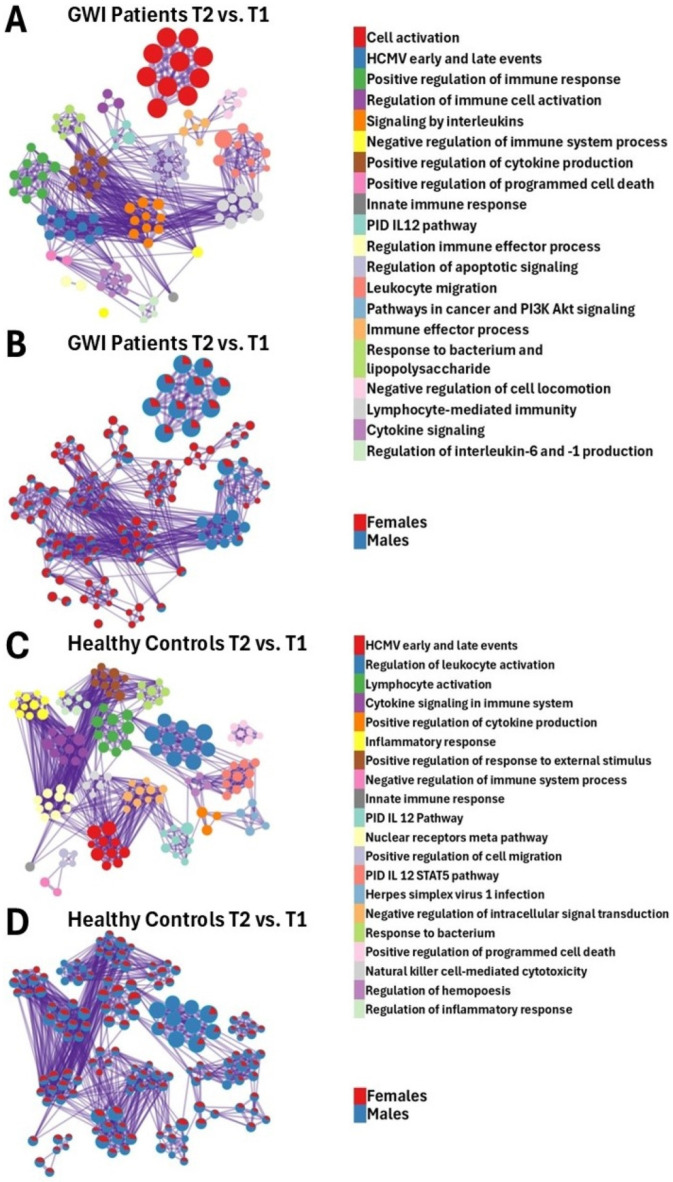
Metascape [18] expression analysis of DEGs between male and female GWI patients (**A**,**B**) and HCs (**C**,**D**) between T2 and T1. Each node represents an enriched term (**A**,**C**). The larger the representation of color, the more enriched it is in each cohort. Cutoff values included a *p*-value < 0.01, a minimum count of 3 in the list of differentially expressed genes, and an enrichment factor of >1.5. GO biological processes, KEGG pathways, Reactome gene sets, CORUM complexes, and canonical pathways from MSigDB were included in the search; (**B**,**D**) nodes are colored by ratios between male (blue) and female (red) participants. The size of the nodes is proportional to the z-score calculated by Metascape [18].

**Figure 4 ijms-26-03610-f004:**
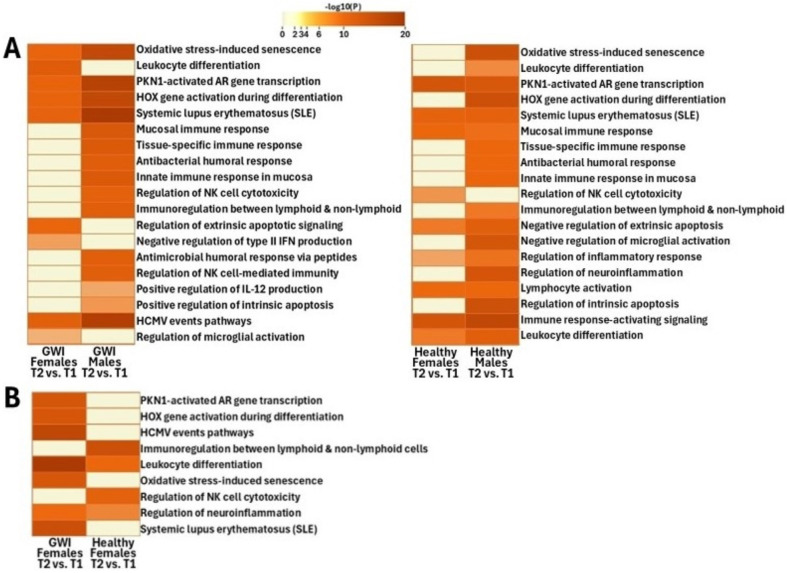
(**A**) Heatmap of the individual pathways that showed the largest difference between T2 and T1 in PBMCs of male and female participants that were found to match those pathways discovered in HCs or exhibited a change in which the cohort was affected compared to HCs. (**B**) A comparison of enriched pathways between female GWI veterans and healthy controls.

**Table 1 ijms-26-03610-t001:** Demographic information comparing female GWI subjects and HCs, including SF-36 questionnaire data. Data are shown as mean ± standard error of the mean, * *p* ≤ 0.05, Student’s *t*-test.

	Category	GWI Females	Healthy Controls	*p*-Value
	Age	53.2 ± 1.4	49.8 ± 1.4	0.09
	BMI	29.0 ± 1.0	26.6 ± 1.2	0.16
Physical Health				
	Physical Function	49.0 ± 4.8	96.3 ± 1.5	<0.00 *
	Role–Physical	20.0 ± 6.6	98.4 ± 1.5	<0.00 *
	Body Pain	33.6 ± 4.0	90.7 ± 2.9	<0.00 *
	General Health	37.7 ± 4.4	79.0 ± 4.6	<0.00 *
Mental Health				
	Vitality	32.8 ± 4.1	65.7 ± 6.3	<0.00 *
	Social Function	24.9 ± 5.0	95.3 ± 2.2	<0.00 *
	Role–Emotional	36 ± 8.5	95.8 ± 2.8	<0.00 *

## Data Availability

Raw data have been submitted to GEO with the accession number GSE286345 and are available to reviewers with the token “evmjucskrbobfwp”.

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
