# Peer review of "Gulf War Illness Induced Sex-Specific Transcriptional Differences Under Stressful Conditions"

_ijms, 2025, doi:10.3390/ijms26083610_

Round 1
Reviewer 1 Report
Comments and Suggestions for Authors
The Frank et al. manuscript focuses on Gulf War illness (GWI). The aim is to uncover potential sex-specific transcriptional differences under stressful conditions in GWI subjects and healthy controls. The authors describe sex-specific differences both in baseline, response to stress and also during recovery from stress. The strength of the study is the unbiased genome-wide transcriptome approach, inclusion of several key criteria and comparison in analysis. In particular, the significance of the study comes from the performed detailed comparisons to an earlier similarly performed study in males. Detailed statistically robust transcriptome analysis was followed up with targeted relevant comparison to published studies. The study is described in adequate detail in the methods section, and the results are discussed in detail in the detailed discussion section with relevant references cited.
Additional notes:
Line 24: “at T1compared’ text in abstract needs to be corrected.
Line 154: Clarify the statement with the number of DEGs, as two different numbers were listed here: “..585 DEGs were found. Of these 584 DEGs,…” (Based on Table S4, number of DEGs are 584.)
Line 198 to 201: The Nanostring validation result section is somewhat underdeveloped. It refers to the data shown in Table S5 for evidence, but not clear what is included in Table S5, and how the “95 percent” conclusion (stated in Line 199) was concluded. This result section should be expanded to further describe the findings and criteria for the conclusion.
Section 3.2.6 “Immune dysfunction” in the Discussion section is rather short, should be expanded or removed, or potentially merged with other sections above describing related terms.
Author Response
We sincerely appreciate the time and effort you have taken to review our manuscript! Your constructive feedback has been very instrumental in refining our work, and we have carefully addressed all your comments and suggestions. Below, we provide point-by-point responses to each of your concerns and outline the changes made in the revised manuscript.
Comment 1: "at T1compared" text in the abstract needs to be corrected
Response 1: Thank you for pointing this out. We have corrected the formatting issue by adding a space to ensure clarity.
Comment 2: Clarify the statement with the number of DEGs, as two different numbers were listed here: “..585 DEGs were found. Of these 584 DEGs,…” (Based on Table S4, number of DEGs are 584.)
Response 2: We appreciate your careful review very much. The correct number is 584, and we have updated the text accordingly to reflect this accurately.
Comment 3: Line 198 to 201: The Nanostring validation result section is somewhat underdeveloped. It refers to the data shown in Table S5 for evidence, but not clear what is included in Table S5, and how the “95 percent” conclusion (stated in Line 199) was concluded. This result section should be expanded to further describe the findings and criteria for the conclusion.
Response 3: We recognize the need for further clarification. This section has been expanded to provide a more detailed explanation of the validation process, and Table S5 has been updated to ensure that the data presentation is clearer.
Comment 4: Section 3.2.6 “Immune dysfunction” in the Discussion section is rather short, should be expanded or removed, or potentially merged with other sections above describing related terms.
Response 4: We appreciate your suggestion. To improve readability and logical flow, we have merged Section 3.2.6 with Section 3.2.5, restructuring the discussion accordingly.
We greatly appreciate your thorough evaluation and constructive recommendations. The revisions have significantly strengthened our manuscript, and we believe the updated version addresses all concerns raised. Thank you for your insightful feedback and for helping us improve the quality of our study.
We want to thank you again for your constructive valuable feedback that has served to strengthen our manuscript.

Reviewer 2 Report
Comments and Suggestions for Authors
Manuscript “Gulf War Illness induced sex-specific transcriptional differences under stressful conditions” presented by Joshua Frank et al. describes an important comparative study of transcriptomes of peripheral blood mononuclear cells collected from Gulf War Illness-affected individuals and healthy controls. Both biological sexes were examined at three-time points involving hard exercise. The main goal was to find potential biomarkers of the disease as well as characterize some aspects of the disease’s mechanisms. Obtained results demonstrated sex-dependent differences in transcriptomes at several time points with a number of potential biomarkers detected. Some of which are known for modulation of immune response and some are new. The work has well well-thought-out experimental design and is presented with sufficient supplementary data. However, there are several critical questions which should be answered before publication.
It is not clear how this work overlaps with the previously published article (https://doi.org/10.1016/j.lfs.2021.119719 ). The authors mentioned that some parts of the used data are already published but what part – only reference to the “Table 1. Demographic information” mentioned? However, 19 male GWI subjects and 25 male HCs are tested in this paper and 19 and 20 corresponding subjects are mentioned in the published paper. At least, there is a discrepancy in HC number. Also, similarities and differences in findings should be explained. For example, Fig. 1. Of the published paper has “GO: 0030335 positive regulation of cell migration” as one of prominent parameters observed during analysis. I, probably, missed this in current publication. HCMV-related parameters difference stated in this study could not be found in previous publication. Also, how HCMV activity relates to studied cells? Are they are collected from people exposed to the virus during the Gulf War? Everything should be clarified.
|
Additional comments (Comments are made during continuous reading of the manuscript. Therefore, answers to some raised questions may occur later in the text.) |
Abstract 1) This sentence is difficult to understand “In female subjects with GWI proinflammatory processes, and in male subjects IL-12 signaling and lymphocytic activation were deregulated at T1compared to T0.”. Please refine. 2) I am not sure that transcriptomics data allow us to make the following conclusions “HCMV activity and microglia activation increased in female GWI subjects, and apoptotic signaling changed in males with GWI.”. There are no direct measurements of these processes. Transcriptomic data can mainly indicate that such differences occur. 3) HCMV activity should be somehow clarified. Is it an endogenous transcript present in cells of all subjects? 4) Healthy controls are not sufficiently described. |
Introduction 1) There is a paragraph describing the methodological aspects of the published work (https://doi.org/10.1016/j.lfs.2021.119719 ). However, this description without mentioning of results and comparison of these results to other relevant publications has almost no value. It is better to elaborate on mentioned “While some advances have been made to determine the underlying mechanisms of GWI”. What are advances? Who achieved this advances? |
Materials and methods 1) Describe in more detail the HC group. Are these people also military veterans/current members? Also, what is the current status of subjects? Do they have similar types of activities e.g. current military members can be engaged in more physically and emotionally active lifestyles than veterans? Physical functioning and physical role functioning are likely to provide related information. However, they are, as I understand, self-reporting parameters. 2) Provide more details on Metascape-based statistical analysis of the data sets. |
Results and discussion 1) What is the reason for much lower enrichment, tanking in account diameter of nodes e.g. Fig. 1-3 (Fig. 3A is a partial exception), in females vs males? In the case of Fig. 1A, female data are looking very sparse. 2) Please add p-values for statements like this “highly significant”. Also, data variability should be demonstrated in the form of graphs, at least for major transcripts. |
Author Response
Thank you so much for taking the time to review our manuscript! We have carefully addressed each of your comments and suggestions and made each revision to enhance the clarity and accuracy of the manuscript. Your feedback has played a crucial role in refining our analysis and strengthening the overall presentation of our findings and below we detail each of our changes.
Comment 1: It is not clear how this work overlaps with the previously published article (https://doi.org/10.1016/j.lfs.2021.119719 ). The authors mentioned that some parts of the used data are already published but what part – only reference to the “Table 1. Demographic information” mentioned?
Response 1: Thank you for bringing this to our attention. In Sections 4.3 and 4.4 we have explicitly described how data from the previous study were incorporated, clarifying both the similarities and differences between the two studies.
Comment 2: 19 male GWI subjects and 25 male HCs are tested in this paper and 19 and 20 corresponding subjects are mentioned in the published paper. At least, there is a discrepancy in HC number.
Response 2: Thank you very much for pointing this out. We acknowledge the inconsistencies and have corrected the numbers to reflect accuracy. These updates ensure consistency with our previous study. Male HCs corrected to 20 (instead of 25) and female HCs corrected to 18 (instead of 17).
Comment 3: Fig. 1. Of the published paper has “GO: 0030335 positive regulation of cell migration” as one of prominent parameters observed during analysis. I, probably, missed this in current publication.
Response 3: We appreciate your valuable feedback regarding differences in the analysis. In our previous published study, this pathway was enriched in male HCs compared to male GWI subjects at the baseline (T0). However, in the current manuscript we focused on comparing transcriptional changes between time points in the exercise challenge, meaning, we compared transcriptomes at T1 time point versus T0 time point and transcriptomes at T2 time point versus transcriptomes at T1 time point in male and female GWI subjects and healthy controls.
Comment 4: How HCMV activity relates to studied cells? Are they are collected from people exposed to the virus during the Gulf War? Everything should be clarified.
Response 4: Thank you for your insightful comment about HCMV activity pathways. Your comments made us look deeper into these pathways and identify differentially expressed genes that made HCMV activity pathways overrepresented. It appears that histone proteins are related to this pathway, and they are part of innate immune response as histones can act as Damage-Associated Molecular Patterns (DAMPs), promoting immune cell activation and pro-inflammatory cytokine release. This correction has been made in Section 3.2.1 to accurately reflect the nature of these findings.
Comment 5: Abstract
1) This sentence is difficult to understand “In female subjects with GWI proinflammatory processes, and in male subjects IL-12 signaling and lymphocytic activation were deregulated at T1compared to T0.”. Please refine.
2) I am not sure that transcriptomics data allow us to make the following conclusions “HCMV activity and microglia activation increased in female GWI subjects, and apoptotic signaling changed in males with GWI.”. There are no direct measurements of these processes. Transcriptomic data can mainly indicate that such differences occur.
3) HCMV activity should be somehow clarified. Is it an endogenous transcript present in cells of all subjects?
Response 5: Thank you very much for pointing this out. We have revised the sentence on inflammatory responses to improve readability. The statement regarding HCMV activity and microglia activation has also been modified to reflect the limitations of transcriptomics data, ensuring it does not overstate conclusions. The nature of HCMV transcripts has been clarified in the Discussion.
Comment 6: It is better to elaborate on mentioned “While some advances have been made to determine the underlying mechanisms of GWI”. What are advances? Who achieved this advances?
Response 6: We appreciate your comments regarding better presentations of published work. We expanded this section in the Introduction and described the main advances in finding underlying molecular mechanisms of GWI.
Comment 7: Describe in more detail the HC group. Are these people also military veterans/current members? Also, what is the current status of subjects? Do they have similar types of activities e.g. current military members can be engaged in more physically and emotionally active lifestyles than veterans? Physical functioning and physical role functioning are likely to provide related information. However, they are, as I understand, self-reporting parameters.
Response 7: We agree that a more precise description is needed. We have now specified that healthy controls were sedentary and not engaged in physically active lifestyles, which could be a relevant factor in their physiological responses.
Comment 8: Provide more details on Metascape-based statistical analysis of the data sets.
Response 8: Thank you for your insightful comment about the Metascape software. We appreciate the opportunity to provide additional information and clarification on how the Metascape statistical analysis was performed, including the parameters used.
Comment 9: What is the reason for much lower enrichment, tanking in account diameter of nodes e.g. Fig. 1-3 (Fig. 3A is a partial exception), in females vs males? In the case of Fig. 1A, female data are looking very sparse.
Response 9: Thank you for your thoughtful comments regarding differences in pathways enrichment between males and females. We have elaborated on possible biological reasons for lower enrichment in females, discussing possible sex-based differences in immune response and transcriptomic variability. However, our study is the first study comparing transcriptomes of male and female GWI subjects, and this is the reason why we tried to be more conservative in our conclusions and not to over speculate. More studies are needed to understand GWI-induced sex specific differences on the level of transcriptomes.
Comment 10: Please add p-values for statements like this “highly significant”.
Response 10: Thank you for pointing this out. We have added p-values to support all statistical claims throughout the manuscript, ensuring transparency and rigor in our findings.
We want to thank you for your thorough review and constructive comments. It helped us to make the manuscript stronger and clearer.

Round 2
Reviewer 2 Report
Comments and Suggestions for Authors
Most of my comments were addressed. Only answer to discrepancies in number of subjects was ambiguous. Was it some kind of typo or entire data set was adjusted? I hope, the authors addressed this comment correctly.
Author Response
Comment: Most of my comments were addressed. Only answer to discrepancies in number of subjects was ambiguous. Was it some kind of typo or entire data set was adjusted? I hope, the authors addressed this comment correctly.
Response: Thank you very much for your very thorough review and catching this discrepancy! It was very helpful.
Indeed, we made mistakes with the numbers of male GWI subjects and female healthy controls. Those were typos that were overlooked when we initially submitted the manuscript. We have 25 female GWI patients, and somehow this number was mistakenly typed for male healthy controls in the initial draft. The dataset was NOT adjusted.
We apologize for lack of attention to details.
